# Identification and Quantification of Coumarins by UHPLC-MS in *Arabidopsis thaliana* Natural Populations

**DOI:** 10.3390/molecules26061804

**Published:** 2021-03-23

**Authors:** Izabela Perkowska, Joanna Siwinska, Alexandre Olry, Jérémy Grosjean, Alain Hehn, Frédéric Bourgaud, Ewa Lojkowska, Anna Ihnatowicz

**Affiliations:** 1Intercollegiate Faculty of Biotechnology of University of Gdansk and Medical University of Gdansk, University of Gdansk, Abrahama 58, 80-307 Gdansk, Poland; izabela.perkowska@phdstud.ug.edu.pl (I.P.); siwinskajoanna@gmail.com (J.S.); ewa.lojkowska@biotech.ug.edu.pl (E.L.); 2Université de Lorraine-INRAE, LAE, 54000 Nancy, France; alexandre.olry@univ-lorraine.fr (A.O.); jeremy.grosjean@univ-lorraine.fr (J.G.); alain.hehn@univ-lorraine.fr (A.H.); 3Plant Advanced Technologies, 54500 Vandœuvre-lès-Nancy, France; frederic.bourgaud@plantadvanced.com

**Keywords:** analytical methods, model plant, natural genetic variation, natural products, simple coumarins

## Abstract

Coumarins are phytochemicals occurring in the plant kingdom, which biosynthesis is induced under various stress factors. They belong to the wide class of specialized metabolites well known for their beneficial properties. Due to their high and wide biological activities, coumarins are important not only for the survival of plants in changing environmental conditions, but are of great importance in the pharmaceutical industry and are an active source for drug development. The identification of coumarins from natural sources has been reported for different plant species including a model plant *Arabidopsis thaliana*. In our previous work, we demonstrated a presence of naturally occurring intraspecies variation in the concentrations of scopoletin and its glycoside, scopolin, the major coumarins accumulating in Arabidopsis roots. Here, we expanded this work by examining a larger group of 28 Arabidopsis natural populations (called accessions) and by extracting and analysing coumarins from two different types of tissues–roots and leaves. In the current work, by quantifying the coumarin content in plant extracts with ultra-high-performance liquid chromatography coupled with a mass spectrometry analysis (UHPLC-MS), we detected a significant natural variation in the content of simple coumarins like scopoletin, umbelliferone and esculetin together with their glycosides: scopolin, skimmin and esculin, respectively. Increasing our knowledge of coumarin accumulation in Arabidopsis natural populations, might be beneficial for the future discovery of physiological mechanisms of action of various alleles involved in their biosynthesis. A better understanding of biosynthetic pathways of biologically active compounds is the prerequisite step in undertaking a metabolic engineering research.

## 1. Introduction

Coumarins are secondary metabolites widely distributed throughout the plant kingdom. They are synthetized via the phenylpropanoid biosynthesis pathway. We can distinguish several simple coumarins like coumarin, scopoletin (7-hydroxy-6-methoxycoumarin), esculetin (6,7-dihydroxycoumarin), umbelliferone (7-hydroxycoumarin), fraxetin (7,8-dihydroxy-6-methoxycoumarin), sideretin (5,7,8-trihydroxy-6-methoxycoumarin) and their respective glycosylated forms–scopolin, esculetin, skimmin, fraxin and sideretin-glycoside, respectively [1]. Figure 1 presents the semi-developed formula of simple coumarins and their glycosides derivatives identified in this research.

Coumarins have been recognized for many years as an important class of pharmacologically active compounds. They have anticoagulant, anticancer, antiviral and anti-inflammatory properties [2]. In addition to the listed medicinal benefits, it was shown recently in numerous studies that coumarins play an important role in iron (Fe) homeostasis, oxidative stress response, plant-microbe interactions and that they can act as signalling molecules in plants [3,4,5,6,7,8]. In the last few years, an increasing number of reports concern the analysis of root extracts and root exudates that are rich in phenolic compounds, such as simple coumarins, which mediate multiple interactions in the rhizosphere. Coumarins were shown to have a strong impact on the plant interactions with microorganisms and play a crucial role in nutrient acquisition [6,9,10,11,12,13,14,15,16,17,18]. Moreover, the root-secreted scopoletin was proved to exert a selective antimicrobial action in the rhizosphere [8]. These numerous reports examining the biochemical and physiological functions of coumarins, make this class of specialized metabolites extremely interesting from a scientific and commercial point of view. The vast majority of these studies were performed using a reference accession, Col-0, of the model plant *Arabidopsis thaliana* (hereinafter Arabidopsis), and its mutants defective at various steps of coumarin biosynthesis.

Here, we conducted the qualitative and quantitative assessment of coumarin content in leaf and root tissue of a set of Arabidopsis natural populations (accessions). Numerous studies on primary and specialized metabolites profiling were conducted using the Arabidopsis model system [19,20,21,22,23,24,25,26,27,28]. Previously, due to the importance of coumarins for human health, most research on their metabolic profiling were carried out on plants of economic importance, such as e.g., sweet potato (*Ipomoea batatas* L.), rue (*Ruta graveolens* L.) or lettuce (*Lactuca sativa* L.) [29,30,31,32,33,34]. One of the first metabolic profiling of root exudates using Arabidopsis natural populations (Col-0, C24, Cvi-0, Ler) was made by Micallef et al. [35] who attempted to correlate them with the compositions of rhizobacterial communities. However, the authors of this work did not undertake the qualitative and quantitative evaluation of the isolated compounds. Consequently, they could only conclude that there are differences between Arabidopsis accessions in terms of the quality and quantity of released substances, which may have an impact on the composition of the rhizobacterial communities.

So far, only a few more studies focusing on the accumulation of coumarins in Arabidopsis natural populations have been published. As shown by our group [36], a significant natural variation in the accumulation of coumarins is present among the roots of Arabidopsis accessions. Using HPLC and GC-MS analytical methods, we identified and quantified coumarins in the roots of selected seven accessions-Antwerpen (An-1, Belgium), Columbia (Col-0, Germany), Estland (Est-1, Estonia), Kashmir (Kas-2, India), Kondara (Kond, Tajikistan), Landsberg *erecta* (Ler, Poland) and Tsu (Tsu-1, Japan). Subsequently, we conducted a QTL mapping and identified new loci possibly underlying the observed variation in scopoletin and scopolin accumulation. Thereby, we demonstrated that Arabidopsis natural variation is an attractive tool for elucidating the basis of coumarin biosynthesis. Other studies focusing on differential accumulation of coumarins between Arabidopsis accessions were conducted by Mönchgesang et al. [14]. A non-targeted metabolite profiling of root exudates revealed the existence of distinct metabolic phenotypes for 19 Arabidopsis accessions. Scopoletin and its glycosides were among phenylpropanoids that differed in the exudates of tested accessions. This research group also focused on the plant-to-plant variability in root metabolite profiles of 19 Arabidopsis accessions [15]. In the current study, a larger set of 28 accessions was chosen, that represent a wide genetic variation existing in Arabidopsis. To increase the scope of this work, we extracted and quantified coumarins from two different types of tissue–roots and leaves. The latter one, in the light of our best knowledge, have never been tested for coumarin content using Arabidopsis natural variation. We believe our results will be beneficial for further studies focusing on a better understanding of coumarin physiological functions and the exact role of enzymes involved in their biosynthesis.

## 2. Results

### 2.1. UHPLC-MS Targeted Metabolite Profiling of Root and Leaf Tissues Reveals Distinct Metabolic Phenotypes for 28 Arabidopsis Accessions

The average content of each tested coumarins (Table 1) grouped by the 28 Arabidopsis accessions (Table 2), type of tissue (extracts from roots and leaves) and method of preparing extracts for analysis (without and after hydrolysis) were depicted through a general heatmap. For each compound, we quantified both the non-glycosylated coumarins—scopoletin (Figure 2A), umbelliferone (Figure 3A), esculetin (Figure 4A), and their respective glycosylated forms—scopolin (Figure 2B), skimmin (Figure 3B), and esculin (Figure 4B), respectively. The concentration µM was based on the fresh weight (FW).

Our analyses made evidence a significant variation in accumulation of all tested compounds between Arabidopsis accessions, both in roots and leaves. In accordance with the current state of knowledge [6,8,36,37,38,39], we identified the coumarin scopoletin and its glycoside, scopolin, to be the major metabolites that accumulate in Arabidopsis roots (Figure 2A,B).

Scopoletin was the most abundant compound in each of the 28 Arabidopsis accessions studied, especially in the roots (from 2.61 to 151.90 µM), but interestingly this phytochemical was also detected in the leaf extracts (Appendix A). The highest amount of scopoletin was detected in Bay-0, Br-0 and Kondara, respectively (Figure 2A), in samples prepared from the roots and subjected to hydrolysis. In non-hydrolyzed root samples, the highest content of scopoletin was detected for the same accessions–Kondara, Bay-0 and Br-0. As expected, scopoletin content in the leaf extracts was several dozen times smaller (from 0.03 to 2.61 µM) when compared with extracts prepared from the root tissue. Amount of scopoletin in the leaf sample was the highest in Br-0, Est-1 and Bay-0 when subjected to hydrolysis and in Est-1, Br-0, Col-0 and Bay-0 when not hydrolyzed.

Relatively large amounts of scopolin (from 2.94 to 67.26 µM) were found in almost all root extracts that were not subjected to hydrolysis (Figure 2B), the highest in Br-0, Fei-0, Ga-0, Leb 3/4, Ri-0, C24 and Bay-0 accessions (Appendix A). We also identified some accessions (Fuk-1, Bay-0, Ri-0, Sha-1 and Eri-1) with relatively high content of scopolin in root extracts after hydrolysis (from 2.04 to 8.09 µM), most probably due to non-effective enzymatic reaction. Interestingly, another set of accessions (Est-1, Sorbo, Bay-0, Kyo-0, No-0, Ga-0, Fei-0, Ws-0) with relatively high scopolin concentration (from 2.58 to 6.48 µM) was also detected in leaf extracts not subjected to hydrolysis. As could be expected, in hydrolysed leaf samples in which sugar residues were cut off and most of scopolin was transformed into scopoletin, the amounts of scopolin were quite low or close to the LOQ.

Interestingly, in this study, we identified small amounts of umbelliferone (from 0.02 to 1.64 µM) in Arabidopsis plants (Figure 3A). Importantly, we detected this phytochemical in all of the hydrolyzed root extracts (Appendix A). The highest levels of umbelliferone were found in Bay-0, Ri-0, Est-1, Col-0, Br-0, C24, Sorbo, Fuk-1 and Leb 3/4, respectively. The quantity of umbelliferone in all leaf extract samples was below LOQ.

Skimmin, which is a glucoside of umbelliferone, was detected and quantified (from 0.69 to 19.80 µM) mostly in root samples of Arabidopsis accessions that were not subjected to enzymatic hydrolysis (Figure 3B). The highest levels were detected in extracts originating from Ga-0, C24, Ws-0, Ri-0, Est-1, Kyo-0 and Eri-0 accessions. It should be noted that skimmin could also be quantified (concentration from 0.04 to 18.76 µM) in all hydrolyzed root extract samples (Appendix A), which needs further investigation.

Most of the results obtained for the leaf tissues were very low and near the LOQ, however in Eri-1, An-1, C24, Col-0, Van-0, Kondara, Ws-0, Ga-0, Fuk-1, Can-0 and Tsu-1 accessions, we observed values slightly above the limit.

Small amounts of esculetin were detected only in a few of root extracts (max. concentration 0.29 µM) and leaf samples (max. concentration 0.16 µM) (Figure 4A). In root non-hydrolysed samples, esculetin was present in Bay-0, Br-0 and Can-0 accessions, while in hydrolysed extracts it was detected in Can-0, Bay-0, Col-0, Ri-0 and Tsu-1 (Appendix A). It may be puzzling that in some accessions, esculetin was only detected in samples which were not subjected to hydrolysis but not in the hydrolysed ones. This is the case for the root extract of Br-0 (0.07 µM), and leaf samples of C24, Br-0, An-1, Col-0, No-0, Ws-0 and Ri-0 accessions (from 0.01 to 0.16 µM, Appendix A). In leaf samples after hydrolysis, only trace amount of esculetin was detected in Ws-0.

Esculin, which is a glycoside form of esculetin, was not found in any root extract (Figure 4B), except Col-0 sample with quantity near to LOQ (0.01 µM). Trace amounts of esculin were detected in some leaf extracts without hydrolysis (from 0.01 to 0.15 µM) with the highest content in Col-0 accession, and in the leaf samples subjected to hydrolysis (from 0.01 to 0.36 uM). Here, the highest esculin content was detected in Ws-0 accession (Appendix A).

### 2.2. Principal Component Analysis (PCA) for 28 Arabidopsis Accessions Using Coumarin Quantification by UHPLC-MS in Selected Geographic and under Diverse Climatic Factors

In order to compare and visualize the possible relationship between coumarin content variability present among 28 Arabidopsis accessions in selected geographic and in various climatic factors (maximal altitude [m], average winter minimal temperature [°C], average summer maximal temperature [°C] and average annual precipitation [mm]), we performed Principal Component Analysis (PCA). About half of the variance of used dataset was covered by the first two principal components, explaining 49% of the overall data variance (27.1% and 21.9% for PC1 and PC2, respectively) (Figure 5A).

According to the results presented on the Variables-PCA plot (Figure 5B), we assumed that there is a positive correlation between scopoletin, umbelliferone and scopolin concentration in root samples before hydrolysis. Despite the fact that scopolin content has relatively small contribution in explaining the variability between tested accessions, it can be also positively correlated with skimmin and umbelliferone concentration. Skimmin content is positively correlated with annual precipitation data.

A negative correlation is highlighted between the following variables: (1) umbelliferone concentration and temperatures (average winter minimal temperature and average summer maximal temperature); (2) scopolin concentration and temperatures (average winter minimal temperature and average summer maximal temperature); (3) skimmin concentration and average summer maximal temperature, as well as skimmin concentration and maximal altitude; (4) scopoletin concentration and annual precipitation, as well as scopoletin concentration and average winter minimal temperature (Figure 5B).

On the Figure 5C we demonstrate the graph of variables (scree plot) which indicates the percentage of variability explained by each dimension (PC). Principal Component 1 and 2 explain 27.1% and 21.9% of the variance respectively, while the other 6 dimensions account for the total remaining variability between each accession (PC3 = 14.7%, PC4 = 13.8%, PC5 = 9.4%, PC6 = 6.2%, PC7 = 3.8% and PC8 = 3%).

## 3. Discussion

Our previous study strongly suggest that Arabidopsis is an excellent model for elucidating the basis of natural variation in coumarin accumulation [36]. Here, we identified and quantified a set of coumarin compounds in the root and leaf methanol extracts prepared from 28 Arabidopsis accessions. In the light of our best knowledge, it is the largest set of Arabidopsis natural populations used in the coumarin profiling analysis that should well represent a wide genetic variation existing in this model plant. It is assumed that these accessions reflect genetic adaptation to local environmental factors [40]. As a result of evolutionary pressure differentially acting on the studied accessions originating from various geographical locations, a large number of genetic polymorphisms is present that have led to different levels of expression of genes involved in the biosynthesis, transport and metabolism of coumarins, and ultimately to different levels of their accumulation. In the current work, we detected a significant natural variation in the content of simple coumarins present in the root and leaf extracts of 28 Arabidopsis accessions. Among tested compounds, scopoletin and its glycosylated form, scopolin, were the most abundant, which is in line with the current state of knowledge [6,14,15,36,37,38].

The previous study on differential accumulation of coumarins between 19 Arabidopsis accessions belonging to the MAGIC lines characterized by a high genetic variability [14], confirm the hypothesis that the composition of Arabidopsis accessions root exudates is genetically determined. They revealed the existence of distinct root metabolic phenotypes among tested natural populations, including variation in the accumulation of scopoletin and its glycoside. Another study focused on extensive profiling of specialized metabolites in root exudates of Arabidopsis reference accession, Col-0, by non-targeted metabolite profiling using reversed-phase UPLC/ESI-QTOFMS [16]. As many as 103 compounds were detected in exudates of hydroponically grown Col-0 plants. Among them, 42 were identified by authenticated standards, including the following coumarins: esculetin, scopoletin, and their glucosides esculin and scopolin. In addition to these coumarins, further esculetin and scopoletin conjugates were initially identified in the root exudates based on their mass spectral fragmentation pattern [16].

It has to be noted that among other coumarin compounds identified in our study, we detected umbelliferone for the first time in Arabidopsis model plant. Authors of the first publication on the accumulation of coumarins in Arabidopsis [37], in which various type of tissues (roots, shoots and callus) were tested, detected trace amounts of skimmin (umbelliferone glucoside) in the wild type plants and slightly increased skimmin level in mutants of CYP98A3. No umbelliferone was detected in that study, or in any other work with Arabidopsis to date, in the light of our best knowledge. It cannot be excluded that we were able to detect umbelliferone due to the sample types tested. We conducted coumarin profiling of extracts prepared from the plant tissues grown in in vitro liquid cultures. Moreover, umbelliferone was detected in root methanol extracts additionally subjected to enzymatic hydrolysis prior to quantification done by UHPLC-MS in order to hydrolyze the glycoside forms of coumarins, while its glycosylated form, skimmin, was detected in samples without enzymatic treatment. It should be highlighted that low amounts of umbelliferone were also detected in non-hydrolysed extracts.

Coumarins have become important players both in optimizing Fe uptake and shaping the root microbiome, thus affecting plant health [5,41]. The link between plant specialized metabolites, in particular coumarins, nutrient deficiencies and microbiome composition that was discovered in recent studies [7,8,42,43], could provide a new set of tools for rationally manipulating the plant microbiome [44]. The selection of underground tissue was an obvious choice in such analyses, considering that coumarins are essentially synthesized in roots where optimization of Fe uptake is coordinated with plant requirements and interaction with soil microorganisms. Therefore, most of the previous coumarin metabolic profiling analysis, including functional characterization of Arabidopsis mutants defective in genes encoding enzymes involved in coumarin biosynthesis or transport, were performed using the root exudates and root tissue [3,6,12,17,38,44,45]. It was also the case in research conducted on the effects of Fe, phosphorus (Pi) or both deficiencies on coumarin profiles in the root tissue of several T-DNA insertional mutants defective in genes involved in Pi or Fe homeostasis [11]. Importantly, in the current study we detected variation in accumulation of esculetin and esculin. The latter one was identified in Arabidopsis leaf extracts, both with and without enzymatic hydrolysis. This requires further research and is of particular interest in the light of recent research findings on coumarin cellular localization, trafficking and signalling [5,7]. Coumarins were found to be involved in the plant response to pathogens in aerial tissues [41,46] and proposed to play an important signalling role in bidirectional chemical communication along the microbiome-root-shoot axis [7].

The study of natural variation in coumarin content present among Arabidopsis accessions is a starting point in elucidating direct links between metabolic phenotypes and genotypes. In the presented research, we also checked whether the climatic and geographic data on the regions from which particular accessions originate, are correlated with the concentrations of tested coumarins. The conducted PCA showed a number of positive and negative correlations between climatic factors and coumarin content. Further investigation is needed to draw a more precise conclusion about possible relationship between the accumulation of coumarins and habitat data. Taking into account, the recent studies showing an important role of coumarins in plant interactions with soil microorganisms and nutrient acquisition, a more in-depth analysis, including data on soil parameters at the origin sites of a given accession, would explain the greater variance and give us more information on the potential correlations. It will be beneficial for the future discovery of physiological mechanisms of action of various alleles involved in the coumarin biosynthesis and can help to select biosynthetic enzymes for further metabolic engineering research.

## 4. Materials and Methods

### 4.1. Chemicals and Reagents

The coumarins standards umbelliferone (purity ≥ 99%), coumarin (>99% purity), esculin (≥98% purity) were purchased from Sigma-Aldrich (St. Louis, MO, USA), scopoletin (>95% purity) and esculetin (>98% purity) from Extrasynthese (Genay, France), skimmin (98% purity) from Biopurify Phytochemicals (Chengdu, China), scopolin (>98% purity) from Chemicals Aktin Inc. (Chengdu, China). Stock solutions of each standard at a concentration of 10 mmol/L were prepared by diluting the powder in dimethyl sulfoxide (Fisher scientific, Illkirch, France) and kept at −18 °C until use. HPLC-grade methanol was purchased from CarloErba Reagents (Val de Reuil, France), formic acid was purchased from Fisher Scientific (Illkirch, France). Water was purified by a PURELAB Ultra system (Veolia Water S.T.I., Antony, France).

### 4.2. Plant Material

All seeds of the 28 Arabidopsis accessions (Table 2) from various habitats which were used in this study were obtained courtesy of prof. Maarten Koornneef.

### 4.3. In Vitro Plant Culture

All the Arabidopsis accessions seeds were surface sterilized with 70% ethanol for 2 min, 5% calcium hypochlorite solution for 8 min and then washed 3 times with sterile ultrapure water. The seeds were placed in Petri dishes with ½ Murashige-Skoog (MS) medium solidified with agar (Sigma-Aldrich) for in vitro plant culture and incubated for 72 h in the dark at 4 °C. Then the plates were transferred to a growth chamber (daily cycle: 16 h light 35 μmol m^−2^ s ^−1^ temperature 20 °C and 8 h dark temperature 18 °C) for 10 days. After that time, seedlings were transferred from agar plates into 200 mL flasks (three individuals per flask) containing 5 mL of ½ MS liquid medium containing 1% sucrose, MS salts, 100 mg/L myo-inositol, 1 mg/L thiamine hydrochloride, 0.5 mg/L pyridoxine hydrochloride and 0.5 mg/L nicotinic acid (Sigma-Aldrich/Merck KGaA, Darmstadt, Germany). Plants were grown in the growth chamber on a rotary platform with shaking 120 rpm. After one week, 3 mL of fresh ½ MS medium was added. Plants were grown for 17 supplementary days and after that time were rinsed with demineralized water, dried on paper towels. Roots and leaves samples were weighted (50 ± 2 mg fresh weight (FW)) and frozen in liquid nitrogen. The plant material was stored in a freezer at −80 °C until extraction process. All accessions were grown in three biological replicates (in three independent flasks, three seedlings per flask).

### 4.4. Metabolites Extraction

For the metabolites extraction, plant tissue frozen in liquid nitrogen was grinded by the usage of 5 mm diameter stainless steel beads (Qiagen, Hilden, Germany). To the 2 mL microtubes, 2 clean beads were added and samples were frozen in liquid nitrogen. Then, using vortex, samples were mixed. For the better performance, the freezing and vortexing procedure was repeated several times until all tissue was powdered. To the powdered tissues 0.5 mL of 80% methanol containing 5 µM 4-methylumbelliferone as an internal standard was added. After that, samples were sonicated for 30 min with ultrasonic cleaner (Proclean 3.0DSP, Ulsonix, Expondo, Berlin, Germany) (70% frequency, sweep function) and incubated in 4 °C in darkness for 24 h. Next day, all samples were vortexed, centrifuged at 13,000× *g* for 10 min and the supernatant was transferred into new microtubes. Centrifugation was repeated in order to get rid of any sediment. The extracts were firstly dried for 2 h in an incubator at 45 °C and then, for the next 2 h in a vacuum centrifuge (Savant SpeedVac vacuum concentrator, Thermo Fisher Scientific, Waltham, MA, USA). To the dried extracts 100 µL of 80% methanol was added to dissolve samples during the night at 4 °C. Then the extracts were vortexed for 10 min and separated by 50 µL. One of the replicates was subjected to enzymatic hydrolysis, and the second one was stored at −20 °C until UHPLC-MS analysis (Shimadzu Corp., Kyoto, Japan).

### 4.5. Enzymatic Hydrolysis

The enzymatic hydrolysis was performed according to Nguyen et al. [47]. Methanolic extracts were subjected to enzymatic hydrolysis with a β-glucosidase (Fluka Chemie GmbH, Buchs, Switzerland) in 0.1 M acetate buffer at a concentration of 0.5 mg/mL in order to determine the amounts of glycosylated compounds (o-glycosides). 50 μL of acetate buffer with β-glucosidase at pH 5.0 (0.1 M sodium acetate, 0.1 M acetic acid and 0.5 mg/mL β-glucosidase buffer) was added to 50 μL of the prepared extract and incubated for 22 h at 37 °C. The reaction was stopped by adding 100 µL of 96% ethanol to the reaction mixture. The extracts were dried in an incubator at 45 °C for 2 h and then for about 1 h in a vacuum centrifuge (Savant SpeedVac vacuum concentrator). The obtained extract was dissolved in 50 µL of 80% methanol overnight and stored at −20 °C until UHPLC-MS analysis.

### 4.6. UHPLC Separation

The coumarins analyses were performed using a NEXERA UPLC-MS system (Shimadzu Corp., Kyoto, Japan) equipped with two UHPLC pumps (LC-30AD), an automatic sampler (SIL-30AC), a photodiode array detector (PDA, SPDM-20A) and combined with a mass spectrometer (single quadrupole, LCMS-2020). Coumarins separation was done on a C18 reversed phase column (ZORBAX Eclipse Plus), 150 × 2.1 mm, 1.8 μm (Agilent Technologies, Santa Clara, CA, USA) protected with an Agilent Technologies 1290 Infinity filter. The column was kept at 40 °C in a column oven (Shimadzu CTO-20AC). Mobile phase consisted of 0.1% formic acid in ultrapure water (buffer A) and 0.1% formic acid in methanol (buffer B) at a constant flow rate of 200 µL/min. The linear gradient solvent system was set as follows: 0 min, 10% B; 16 min, 70% B; 18 min, 99% B; 18.01 min, 10% B; 20 min, 10% B. The total analysis duration was 20 min. The injection volume was 5 μL.

### 4.7. MS Detection

The UHPLC system was connected to the MS by an electrospray ionization source (ESI), operating in positive mode (ESI+) and scanning in single ion monitoring mode (SIM). The inlet, desolvation line and heating block temperatures were set at 350 °C, 250 °C, and 400 °C, respectively. The capillary voltage was set at 4.5 kV. Dry gas flow was set at 15 L/min and nebulizing gas at 1.5 L/min. The instrument was operated and data were processed using LabSolution software version 5.52 sp2 (Shimadzu Corp., Kyoto, Japan).

### 4.8. Peak Identification and Quantitation

Each standard molecule was individually injected in the UHPLC-MS in full scan mode to determine retention time and *m/z* ratio for the analysis. The quantitation of each molecule (Table 2) was based on the signal obtained from the MS detection, using angelicin, as an analytical internal standard. Angelicin was added at the same concentration (5 μM) in all the samples before injection as well as in 7 calibration solutions. The calibration solutions contained all of the standard molecules at the same concentrations ranging from 0.1 to 10 μM (0.1, 0.2, 0.5, 1, 2, 5 and 10 μM). Calibration curves were drawn for each compound by linking its relative peak area (compound area divided by the angelicin area) and its concentration. Each curve fit type was linear. The limit of quantitation (LOQ) was calculated as the analyte concentration giving signal to signal to noise ratios (S/N) of 10. Three measurements were assessed per accession.

### 4.9. Principal Component Analysis (PCA)

Principal Component Analysis (PCA) were performed using *prcomp()* package and visualize with the *factoextra* 1.0.7 version package in the R 4.0.4 environment developed by the R Core Team (2020). R: A language and environment for statistical computing. R Foundation for Statistical Computing, Vienna, Austria (www.R-project.org (accessed on 20 January 2021)) and the RStudio Team (2019). RStudio: Integrated Development for R. RStudio, Inc., Boston, MA, USA (www.rstudio.com (accessed on 20 January 2021)). All the variables were standardized before analysis. Data used for the analysis are presented in Appendix A.

## 5. Conclusions

Multi-pharmacological properties of coumarins that are widely used in medical applications, make the study of coumarin biosynthesis attractive from the commercial point of view. Considering that all medicinal plants currently used in studying the biosynthesis of coumarins are non-model organisms and many approaches are not available in those species, makes a model plant Arabidopsis, with its extensive genetic variation and numerous publicly accessible web-based databases, an excellent model to study accumulation of coumarins in natural populations. The presented results focusing on qualitative and quantitative characterization of natural resources provide a basis for further research on identification of genetic variants involved in coumarin biosynthesis in plants, which is the first step in metabolic engineering for the production of natural compounds. We identified scopoletin, and its glycosylated form, scopolin, to be the most abundant coumarins in Arabidopsis tissues. It should be emphasized that among other coumarin compounds identified in this study, we detected umbelliferone for the first time in Arabidopsis. In view of the considerable importance of umbelliferone in synthesis and its pharmacological properties, this is a significant step in the study of biosynthesis of coumarins using this model plant.

## Figures and Tables

**Figure 1 molecules-26-01804-f001:**
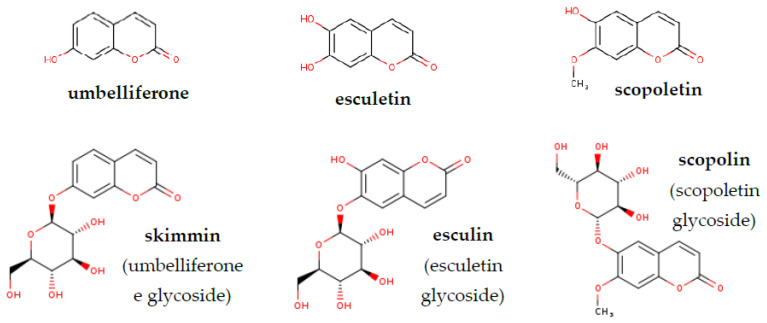
Chemical structures of simple coumarins and their glycosides analysed in this work (www.chem-space.com (accessed on 20 January 2021)).

**Figure 2 molecules-26-01804-f002:**
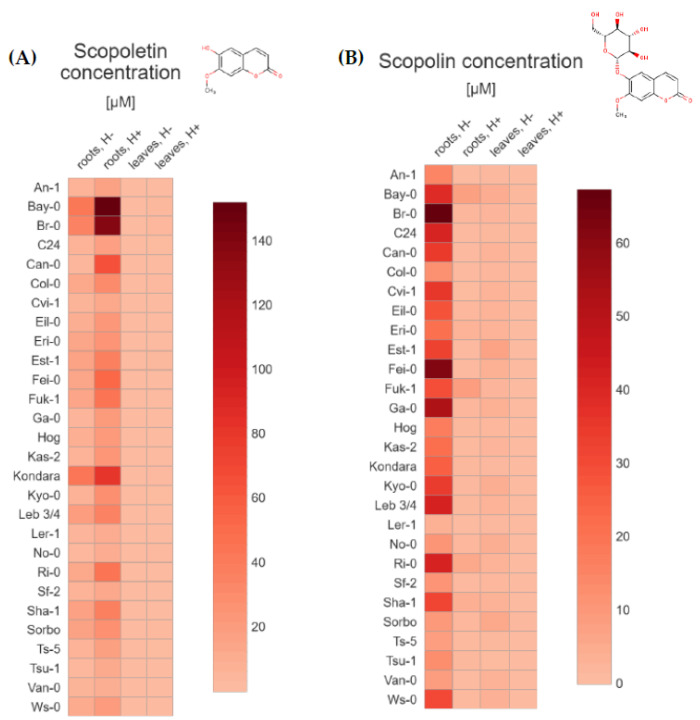
Heat maps based on the average (**A**) scopoletin and (**B**) scopolin concentration (µM/FW) in Arabidopsis tissue extracts from roots and leaves, without and after hydrolysis. The values used in the plots (https://app.displayr.com (accessed on 20 January 2021)) are the mean of 3 biological replicates. The mean values and standard deviations (±SD) are gathered in the Appendix A.

**Figure 3 molecules-26-01804-f003:**
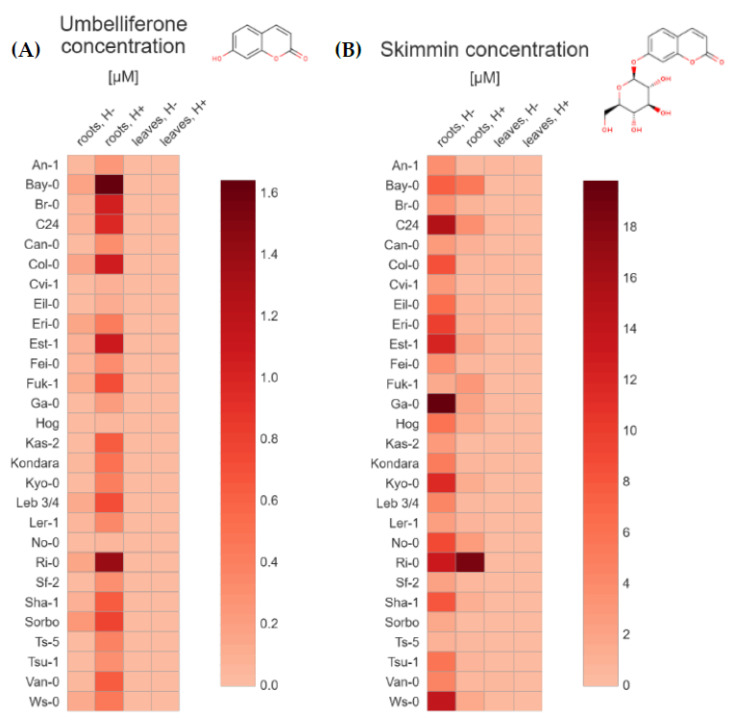
Heat maps based on the average (**A**) umbelliferone and (**B**) skimmin concentration (µM/ FW) in Arabidopsis tissue extracts from roots and leaves, without and after hydrolysis. The values used in the plots (https://app.displayr.com (accessed on 20 January 2021)) are the mean of 3 biological replicates. The mean values and standard deviations (±SD) are gathered in the Appendix A.

**Figure 4 molecules-26-01804-f004:**
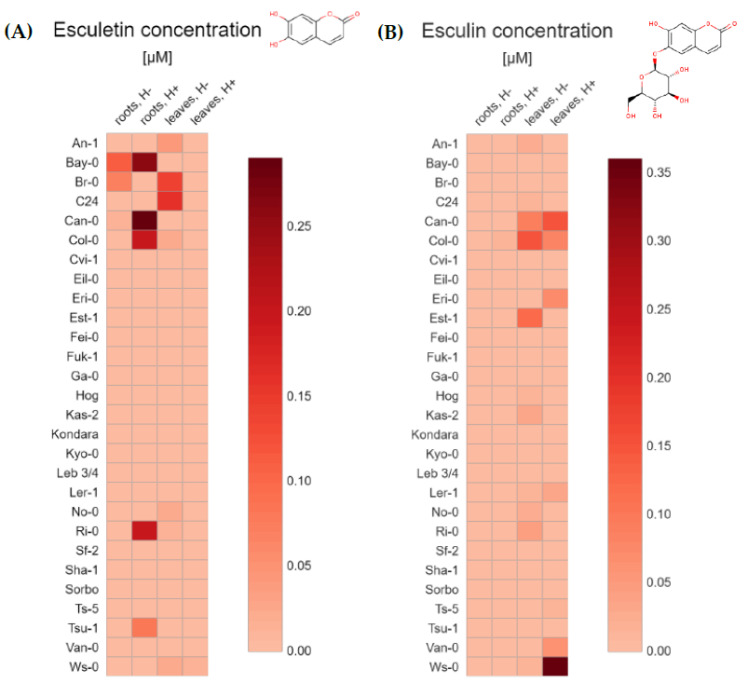
Heat maps based on the average (**A**) esculetin and (**B**) esculin concentration (µM/ FW) in Arabidopsis tissue extracts from roots and leaves, without and after hydrolysis. The values used in the plots (https://app.displayr.com (accessed on 20 January 2021)) are the mean of 3 biological replicates. The mean values and standard deviations (±SD) are gathered in the Appendix A.

**Figure 5 molecules-26-01804-f005:**
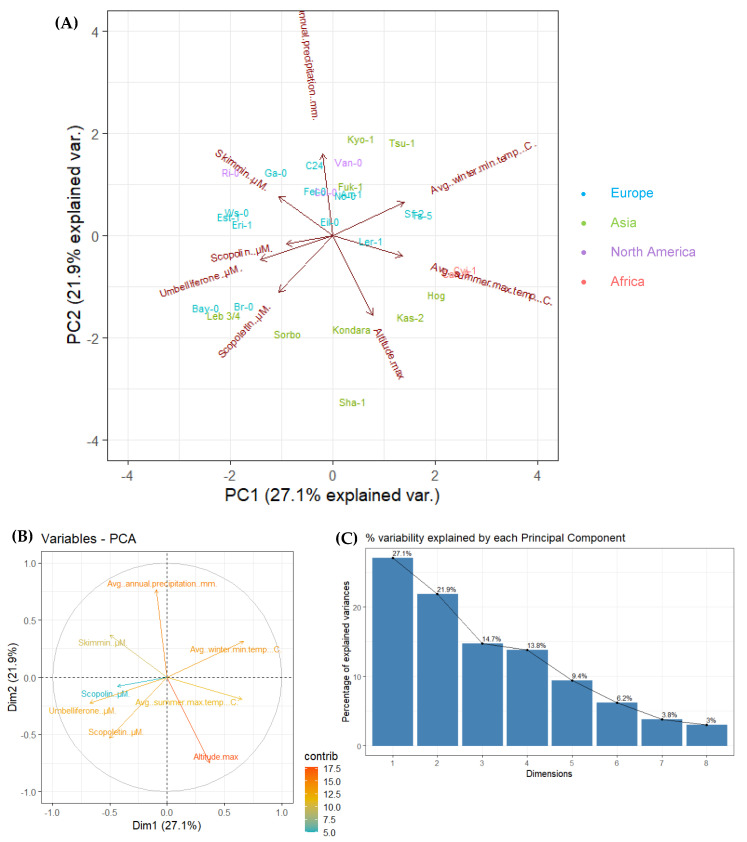
(**A**) Principal component analysis (PCA) for 28 Arabidopsis accessions using the concentration of umbelliferone, scopoletin and their corresponding glycosides (skimmin and scopolin, respectively) in root samples without hydrolysis, and four geographic and climatic factors (maximal altitude [m], average winter minimal temperature [°C], average summer maximal temperature [°C] and average annual precipitation [mm]; Appendix A). Factor coordinates are marked with arrows. Observations indicated by blue accession names represent European locations (n = 14), green represent Asian locations (n = 9), violet represent North American locations (n = 3) and red represent African locations (n = 2). The abbreviations indicate the accessions according to Table 2. Component one and two explain 49% of the point variability. (**B**) The Variables-PCA contribution plot shows the correlation of the variables used in PCA with the respective contribution of each factor (contrib) indicated with a colour gradient. (**C**) The scree plot/graph of variables demonstrate the percentage of variability explained by each dimension (PC). Principal Component 1 and 2 explain 27.1% and 21.9% of the variance respectively.

**Table 1 molecules-26-01804-t001:** Coumarins and their glycosides identified in this study.

Peak Number	Retention Time tR (min)	Compound	Mass (*m*/*z* Ratio)	LOQ
(1)	14.5	Umbelliferone	163 (M+H^+^)	0.2 µM
(2)	11.8	Esculetin	179 (M+H^+^)	0.5 µM
(3)	14.8	Scopoletin	193 (M+H^+^)	0.2 µM
(5)	10.1	Skimmin (glycosylated umbelliferone)	325 (M+H^+^)	0.1 µM
(5)	11.8	Esculin (glycosylated esculetin)	341 (M+H^+^)	0.1 µM
(6)	11	Scopolin (glycosylated scopoletin)	355 (M+H^+^)	0.1 µM

**Table 2 molecules-26-01804-t002:** Basic information on the Arabidopsis accessions used in this study.

No.	Full Name	Abbreviation	Country of Origin
1	Antwerpen	An-1	Belgium
2	Bayreuth	Bay-0	Germany
3	Brunn	Br-0	Czech Republic
4	Coimbra	C24	Portugal
5	Canary Islands	Can-0	Spain
6	Columbia	Col-0	USA
7	Cape Verdi	Cvi-1	Cape Verde Islands
8	Eilenburg	Eil-0	Germany
9	Eringsboda	Eri-1	Sweden
10	Estland	Est-1	Russia
11	St. Maria d. Feiria	Fei-0	Portugal
12	Fukuyama	Fuk-1	Japan
13	Gabelstein	Ga-0	Germany
14	Hodja-Obi-Garm	Hog	Tajikistan
15	Kashmir	Kas-2	India
16	Kondara	Kondara	Tajikistan
17	Kyoto	Kyo-1	Japan
18	Lebjasche	Leb 3/4	Russia
19	Landsberg *erecta*	Ler-1	Germany
20	Nossen	No-0	Germany
21	Richmond	Ri-0	Canada
22	San Feliu	Sf-2	Spain
23	Shakdara	Sha-1	Tajikistan
24	Sorbo	Sorbo	Tajikistan
25	Tossa del Mar	Ts-5	Spain
26	Tsushima	Tsu-1	Japan
27	Vancouver	Van-0	Canada
28	Wassilewskija	Ws-0	Belarus

## Data Availability

The data presented in this study are available within the article and Appendix A.

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
