# Peer review of "Identification and Quantification of Coumarins by UHPLC-MS in Arabidopsis thaliana Natural Populations"

_molecules, 2021, doi:10.3390/molecules26061804_

Round 1
Reviewer 1 Report
The qualitative and quantitative assessment of coumarin content in leaf and root tissue of Arabidopsis thaliana was presented. Although some differences have been proposed, the mechanism behind the differences is still unclear. The paper provides an analytical and quantitative reference for further research.
Some points below:
There are too many paragraphs in the introduction section. Please logically clarify the purpose of the paper and related research progress, 3 or 4 paragraphs are appropriate.
The same in the dissuccion section, please simplify and make the main point.
The concentration µM is based on dry or wet materials, please clarify.
Line 348, Please give the sonicated condition.
Under optimized UPLC/MS conditions, it is helpful to give a representative UPLC or total ion current chromatograms.
Reviewer 2 Report
The article describes the profiling of Arabidopsis thaliana for coumarins. The study is elegantly designed and presented. The results are well discussed with detailed references to the results of other authors. Particularly the discussion is the strong point of this work.
The article can be published after supplementing this information:
Quality control discussion is missing. Have authors used blank samples, reference materials?
In few places of the manuscript it is “principle component analysis” instead of “principal component analysis”.
The results of PCA are not that clear. There is not separation between groups of objects so not geographical grouping can be done with PCA. The presentation of factor loadings would benefit in understanding how PCs are constructed. Also the information on variance explained by other PCs should be given. Maybe the objects are separated by different PCs?
Reviewer 3 Report
This paper reported the extraction and the identification of coumarins from roots and leaves of 28 Arabidopsis accessions. The novelty of this work is satisfactory and provided advancement in scientific knowledge. The paper is well written with good plan of experimental analysis and conclusions can be justified based on the experimental results. To be published this paper needs minor revision:
Some errors through the text, please revise.
For quantification experiments, how many measurements were assessed? Please add.
